# Untied Ulysses: Memory-Efficient Context Parallelism via Headwise Chunking

**Ravi Ghadia**[1] **Maksim Abraham**[1] **Sergei Vorobyov**[1] **Max Ryabinin**[1]

## Abstract

Efficiently processing long sequences with Transformer models usually requires splitting the computations across accelerators via context parallelism. The dominant approaches in this family of methods, such as Ring Attention or DeepSpeed Ulysses, enable scaling over the context dimension but do not focus on memory efficiency, which limits the sequence lengths they can support. More advanced techniques, such as Fully Pipelined Distributed Transformer or activation offloading, can further extend the possible context length at the cost of training throughput. In this paper, we present **UPipe**, a simple yet effective context parallelism technique that performs fine-grained chunking at the attention head level. This technique significantly reduces the activation memory usage of self-attention, breaking the activation memory barrier and unlocking much longer context lengths. Our approach reduces intermediate tensor memory usage in the attention layer by as much as **87.5**% for 32B Transformers, while matching previous context parallelism techniques in training speed. UPipe can support the context length of **5M** tokens when training Llama3-8B on a single 8×H100 node, improving upon prior methods by over **25**%.

## 1. Introduction

The Transformer architecture (Vaswani et al., 2017) has powered significant advances in AI in recent years, ranging from language models with agentic and reasoning capabilities (Gemini Team, 2025; Kimi Team et al., 2025; Chen et al., 2025) to video generation (Team Wan et al., 2025; Wu et al., 2025; Sand.ai et al., 2025). As the field continues to progress, the demand for longer context lengths in AI models continues to grow due to applications such as code

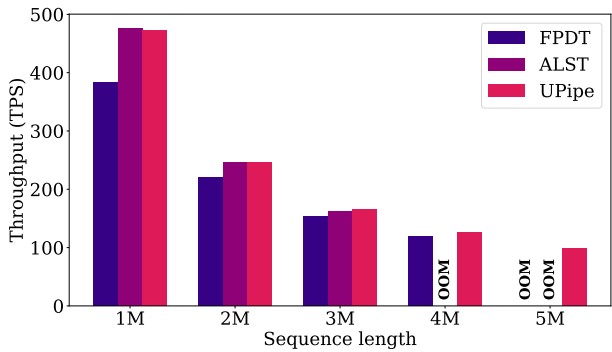

*Figure 1.* Comparison of context parallelism approaches on long-sequence training for Llama3-8B using 8 × H100s. UPipe provides maximum efficiency, resulting in a longer maximum context length (5M tokens) while retaining throughput.

generation (Li et al., 2023a; Hui et al., 2024), long document understanding (Chia et al., 2024; Jiang et al., 2024), or even audio processing (Hori et al., 2021). However, training models to effectively process such long sequences is limited by the accelerator hardware: beyond a certain limit, even keeping the activations necessary for self-attention becomes a bottleneck. As a result, methods that reduce the memory requirements of long-context training have recently attracted a surge of research interest.

The most scalable approaches for increasing the context size beyond a single accelerator leverage distributed training, splitting the computations and memory allocations across multiple devices. In particular, the context parallelism (Li et al., 2022; 2023b; Jacobs et al., 2023) family of methods (also known as sequence parallelism) focuses on sharding model operations across the sequence axis. These methods enable effective scaling in context length with the number of accelerators, but the activation memory per device still scales linearly with the sequence length. Therefore, at very long sequence lengths (>2M), the activation memory starts to become a bottleneck, limiting the training capacity.

In this paper, we propose **UPipe**, a context parallelism method that focuses on improving the memory efficiency of long-context training while maintaining performance on par with current approaches. Our method is designed on the principle that for long-context training, processing a subset of heads at once is enough to saturate the GPU. Therefore, serializing the execution in the attention layer by grouping

[1]Together AI. Correspondence to: Ravi Ghadia <rghadia@utexas.edu>, Max Ryabinin <mryab@together.ai>.

*Proceedings of the 43$^{rd}$ International Conference on Machine Learning*, Seoul, South Korea. PMLR 306, 2026. Copyright 2026 by the author(s).

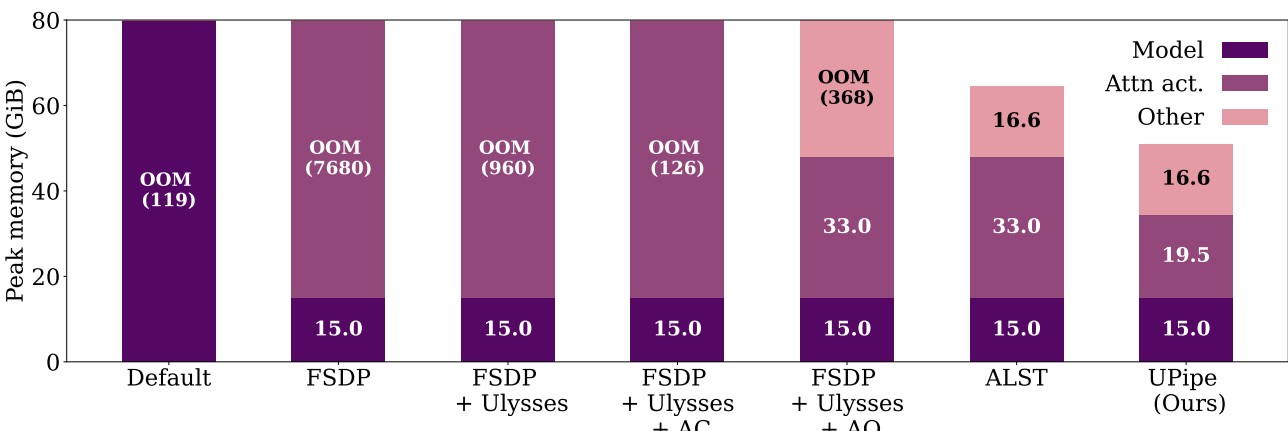

*Figure 2.* Memory usage breakdown when training Llama 3-8B with a sequence length of 3M tokens across 8 H100 GPUs. FSDP denotes Fully Sharded Data Parallel, AC denotes Activation Checkpointing, AO denotes AC with offloading, OOM denotes Out of Memory.

heads into smaller chunks allows for much better memory reuse. UPipe is agnostic to the underlying attention algorithm: similar to DeepSpeed Ulysses, it uses the same kernels to compute attention as non-distributed training.

On Llama 3-8B, our method can fit context lengths of **up to 5 million tokens on a single H100 node**, and up to 8M on two H100 nodes with unified sequence parallelism (Fang & Zhao, 2024), outperforming prior works by **25%** and **33%** respectively in terms of maximum context length supported. At the same time, UPipe's training throughput is comparable to other context parallelism techniques.

Our contributions are as follows:

1. We propose UPipe, a new context parallelism method that executes the attention layer in multiple stages, processing attention heads in chunks. This method is easy to implement and can work as a plug-and-play replacement for existing techniques.

2. We analyze the memory usage of long-context Transformer training, identifying the major bottleneck left unaddressed by prior works and showing how UPipe mitigates this bottleneck.

3. We present a schedule compatible with Grouped-Query Attention (GQA), which processes heads out of order to avoid redundant communication while retaining the memory benefits of standard GQA architecture.

4. We compare UPipe with prior approaches to context-parallel training, measuring the speed and memory usage for 8B and 32B dense Transformer models across sequence lengths ranging from 128K to 5M tokens[1]. UPipe can support longer sequence lengths (up to 5 million tokens on a single 8×H100 node) with negligible performance differences, compared to other methods.

---

[1]The code for our experiments is available at github.com/ghadiaravi13/Untied-Ulysses.

## 2. Background

Long-context Transformer training is increasingly important for use cases such as voice (Yang et al., 2024) and video generation (Sand.ai et al., 2025), but standard self-attention scales poorly with sequence length. For instance, Team Wan et al. (2025) report that the activation memory for training a 14B Diffusion Transformer at a sequence length of 1M tokens reaches 8 TB, far beyond the capacity of a single accelerator and necessitating distributed training.

### 2.1. Multi-head Attention

Multi-head self-attention (Vaswani et al., 2017) is a neural network layer for processing sequential data. For head $h$, each input vector $X_i$ is projected into query $Q_i^h$, key $K_i^h$, and value $V_i^h$ vectors, and the output is computed as softmax $\left( \frac{Q_i^h \cdot K_{\leqslant i}^h}{\sqrt{d_h}} \right) \cdot V_{\leqslant i}^h$, where $d_h$ is the query vector size. Since each query attends to *all* past keys and values, computing attention requires access to the entire sequence, which becomes the key obstacle for long-context training. Grouped-Query Attention (GQA, Ainslie et al., 2023) shares the same key/value heads among groups of $G$ query heads, lowering the key/value memory use by a factor of $G$.

### 2.2. Context Parallelism

To mitigate the large activation memory footprint, Li et al. (2023b) introduced context parallelism in the form of Ring Attention. It shards the context across $C$ devices and exchanges $K, V$ tensors in a ring via peer-to-peer communication. This incurs $\mathcal{O}(C)$ communication calls per attention operation. Later, Liu et al. (2023) proposed a variant of Ring Attention that is compatible with online softmax computation. DeepSpeed-Ulysses (Jacobs et al., 2023), built upon Megatron-SP (Korthikanti et al., 2022), instead rearranges tensors via a single all-to-all collective, reducing communication latency while unlocking optimized self-attention kernels. We discuss this technique in detail in Section 3.1.

*Table 1.* Theoretical peak memory usage breakdown across different stages of the forward pass for a Transformer model. All floating point tensors use `BFloat16` precision by default, except cross-entropy loss, which uses `fp32` precision.

| Stage | Inputs | Intermediate tensors | | | Outputs | Total |
|---|---|---|---|---|---|---|
| | | Type | Memory | Ratio | | |
| ❶ Embedding | $4 \cdot S$ (`int32`) | – | – | – | $2 \cdot S \cdot d_{model}$ | $2 \cdot S \cdot d_{model}$ |
| ❷ Attention | $2 \cdot S \cdot d_{model}$ | QKV all-to-all | $6 \cdot S \cdot H \cdot d_{head}$ $6 \cdot S \cdot H \cdot d_{head}$ | $H = d_{model}/d_{head}$ | $2 \cdot S \cdot d_{model}$ | $16 \cdot S \cdot d_{model}$ |
| ❸ Feed-forward | $2 \cdot S \cdot d_{model}$ | Intermediate | $8 \cdot S \cdot d_{ff}$ | $d_{ff} \approx 2.67 \cdot d_{model}$ | $2 \cdot S \cdot d_{model}$ | $25 \cdot S \cdot d_{model}$ |
| ❹ Cross-entropy | $2 \cdot S \cdot d_{model}$ | Logits + LogSoftmax | $8 \cdot S \cdot V$ | $V \approx 30 \cdot d_{model}$ | Loss | $240 \cdot S \cdot d_{model}$ |

Further works introduce additional optimizations for memory and throughput. For instance, USP (Unified Sequence Parallelism, Fang & Zhao, 2024) uses DeepSpeed-Ulysses within the node and Ring Attention across nodes, running all-to-all communication over the faster NVLink fabric and ring communication over the slower one.

To address the memory bottleneck of very long contexts, other papers have explored tiling or chunking techniques. In particular, Arctic Long Sequence Training (ALST, Bekman et al., 2025) uses tiling for feed-forward layer and cross-entropy loss calculations, lowering the memory pressure of intermediate tensors. However, it does not address the memory overhead in the attention phase. Fully Pipelined Distributed Transformer (FPDT, Yao et al., 2025) addresses the attention memory overhead by chunking computations along the sequence length dimension and offloading to CPU. However, it suffers from reduced performance due to CPU overhead and additional memory transfers. UPipe addresses the memory overhead while maintaining performance and extends to hybrid schemes such as USP. Also, our technique performs chunking along the head dimension, which is orthogonal and complementary to FPDT.

### 2.3. Activation memory for a Transformer model

Next, we quantify and analyze the memory usage in each phase of a long-context Transformer training step. We identify the factors that impose the biggest memory bottlenecks and cover the techniques to address them.

Consider a decoder-only Transformer with $L$ layers, with $H$ heads per layer, and a GQA group size of $G$. Let the model's hidden size be $d_{model}$, per-head dimension $d_{head}$, the intermediate dimension of the feed-forward network $d_{ff}$, and the vocabulary size of the model $V$. Finally, let a sequence of length $S$ be the input to the model. For now, we keep the batch size equal to 1, since our focus is to establish how memory scales with respect to the sequence length $S$.

We assume a mixed precision setup with parameters, activations, and gradients in `bfloat16` precision, thus requiring 2 bytes per parameter for each tensor. However, some intermediate tensors still require 4 bytes, as discussed later.

For a decoder-only Transformer model, the sequence of operations during the forward pass can be broken down into 4 phases, as shown in Table 1.

❶ **Embedding:** The input sequence of tokens is converted into embedding vectors of size $S \cdot d_{model}$, thus requiring $2 \cdot S \cdot d_{model}$ bytes of memory.

❷ **Attention:** The input gets transformed into query $Q$, key $K$ and value $V$ vectors, requiring $6 \cdot S \cdot H \cdot d_{head}$ bytes. As we will discuss in Section 3.1, all-to-all requires the same amount of additional memory. Assuming we use Flash Attention (Dao et al., 2022), the attention computation would not require any additional GPU HBM memory, except for storing the final output (and the final log-sum-exponent) vectors of size $2 \cdot S \cdot d_{model}$ (and $2 \cdot S \cdot H$ respectively). For most Transformer models, $H = d_{model}/d_{head}$; therefore, this phase has a total memory usage of $2 + (6 + 6) + 2 = 16 \cdot S \cdot d_{model}$ bytes.

❸ **Feed-forward:** Next, the attention output is fed to the feed-forward network (FFN) with two layers. Assuming SwiGLU activations (Shazeer, 2020), this module projects the input with dimension $d_{model}$ into four intermediate tensors of dimension $d_{ff}$ and then projects the intermediate tensors into the output of dimension $d_{model}$. Typically, $d_{ff} \approx 2.67 \cdot d_{model}$, which results in the total memory usage of $25 \cdot S \cdot d_{model}$ bytes.

❹ **Cross-entropy loss:** At the end of the final layer, the output is converted into logits of shape $S \cdot V$, which are then used by the cross-entropy loss function. This phase has the most critical memory constraints, because the vocabulary size $V$ is typically very large ($\approx 30 \cdot d_{model}$). Moreover, the cross-entropy calculation requires logits and intermediate log-softmax values to be casted to `fp32`, making the overall memory consumption $240 \cdot S \cdot d_{model}$ bytes.

### 2.4. Mitigating memory overheads

From the above analysis, we can identify the most critical factors that become a memory bottleneck when scaling the context length. Due to its memory usage, the cross-entropy loss is the first obstacle. We can overcome it by computing

*Table 2.* Peak activation memory within the forward attention block under GQA, expressed in units of one local hidden-state shard ($\frac{S}{C} \cdot d_{model}$ `bf16` elements, i.e., $2 \cdot \frac{S}{C} \cdot d_{model}$ bytes). $\pi$ represents the number of chunks in FPDT, $\nu$ represents the number of chunks in UPipe. UPipe consumes $\nu$ times less intermediate (QKV + all-to-all) activation memory than Ulysses with activation offloading. FPDT has lower memory usage due to an arbitrary chunk size, but suffers from performance degradation.

| Method | Before attn block | During inp_all_to_all | During attn kernel | During out_all_to_all |
|---|---|---|---|---|
| Ulysses | $L$ | $L + (\gamma + 1)$ | $L + (\gamma + 1)$ | $L + 2$ |
| Ulysses + offloading | 1 | $\gamma + 2$ | $\gamma + 2$ | 3 |
| FPDT | $\frac{1}{\pi}$ | $\frac{\gamma + 2}{\pi}$ | $\frac{2\gamma + 1}{\pi}$ | $\frac{2}{\pi}$ |
| Untied Ulysses | 1 | $2 + \frac{\gamma}{\nu}$ | $2 + \frac{\gamma}{\nu}$ | $2 + \frac{1}{\nu}$ |

the loss in a tiled manner, materializing the intermediate tensors only one tile at a time. To reduce the memory usage of the FFN layer, we adopt a similar approach to ALST by using a tiled FFN function. Similarly to other prior works and libraries like Unsloth (Han et al., 2023), we manage the activation memory across layers via full activation checkpointing with CPU offloading. Finally, for the attention layer, we propose **UPipe**, described in Section 3.3.

Table 2 provides the memory consumption of different stages of attention during the forward pass. In grouped-query attention, the size of key ($K$) and value ($V$) tensors is reduced by a factor of $G$. We define $\gamma = 1 + \frac{2}{G}$ as the combined size of $Q, K, V$ (relative to $S/C \cdot d_{model}$). Table 7 of Appendix A presents a similar breakdown for the backward pass of an attention layer (with grouped-query attention).

## 3. Untied Ulysses

As discussed above, the majority of activation tensors during training that contribute to peak memory usage can be tiled, recomputed or offloaded. In this section, we outline the design of UPipe, highlighting its key ideas that enable memory savings in the attention stage. Since our method builds on top of DeepSpeed-Ulysses (Jacobs et al., 2023), we will first elaborate on the mechanics of this algorithm, highlighting the opportunity for memory optimization.

### 3.1. DeepSpeed-Ulysses

DeepSpeed-Ulysses (DS-Ulysses) is a context parallelism technique that allows training of Transformer models in a distributed setup by sharding input sequences across devices. For a sequence of length $S$ and $C$ devices, every device will have a sequence of length $S/C$. Token-wise operations (feed-forward layer, RMSNorm, cross-entropy) can be applied to every sequence shard without communication. However, the attention layer requires access to the entire sequence, so it cannot be executed independently on each shard. Ulysses solves this problem by rearranging the shards: the memory occupied by the shards is the same as before, but now every device has access to the full sequence.

Let us walk through how DeepSpeed-Ulysses works, using an example from Figure 3a with $C = 2$ devices. Initially, the

sequence $X$ of size $[S, d_{model}]$ is sharded on two devices: $X_0$ and $X_1$ with size $[\frac{S}{2}, d_{model}]$ each. These shards then undergo the $QKV$ projection to produce the query, key, and value tensors. In this example, let us consider the number of heads $H = 4$, which means the query, key, and value tensors have 4 heads: $H_0, H_1, H_2, H_3$ with $d_{head} = d_{model}/4$. Hence, the size of these tensors is $[\frac{S}{2}, 4, d_{head}]$.

Next, DS-Ulysses performs an all-to-all operation **inp_all_to_all** on the $QKV$ tensors to *reshard* them: switching the sharding from the sequence length dimension to the head dimension. Thus, $QKV$ tensors are resharded from $[\frac{S}{2}, 4, d_{head}]$ to $[S, 2, d_{head}]$. As a result, device 0 now has access to the full sequence for heads $H_0, H_1$, while device 1 has access to the full sequence for heads $H_2$ and $H_3$. After attention, device 0 computes outputs for heads $H_0$ and $H_1$ (and device 1 computes outputs for $H_2$ and $H_3$), which are then sharded back to the original shape. Therefore, another all-to-all operation **out_all_to_all** is performed, so that the final outputs $O_0$ and $O_1$ are of shape $[\frac{S}{2}, d_{model}]$ each.

Because of all-to-all communication, DS-Ulysses can fully utilize the network topology between all context-parallel ranks and does not incur a latency overhead of $\mathcal{O}(C)$ peer-to-peer transfers, which makes it more performant than Ring Attention (Gu et al., 2024). However, it faces a memory bottleneck for long-context training, because the overhead of communication buffers can become significant.

### 3.2. Memory Usage

The memory usage peak in DeepSpeed-Ulysses occurs during the **inp_all_to_all** operation because of the memory buffers for query, key, and value tensors and similarly sized all-to-all buffers. From Table 1, we can see that the memory usage due to these intermediate tensors is proportional to $H$. Our key insight is that *for long sequences and large enough models, performing attention over a subset of heads is enough to reach the compute-bound regime*. Hence, we process only $U$ heads at a time, where $U < H$. Because the memory usage of intermediate tensors is proportional to the number of heads, the memory overhead reduces from $\mathcal{O}(H)$ to $\mathcal{O}(U)$. As we show later, changing the hyperparameter $U$ allows for a natural runtime-memory tradeoff.

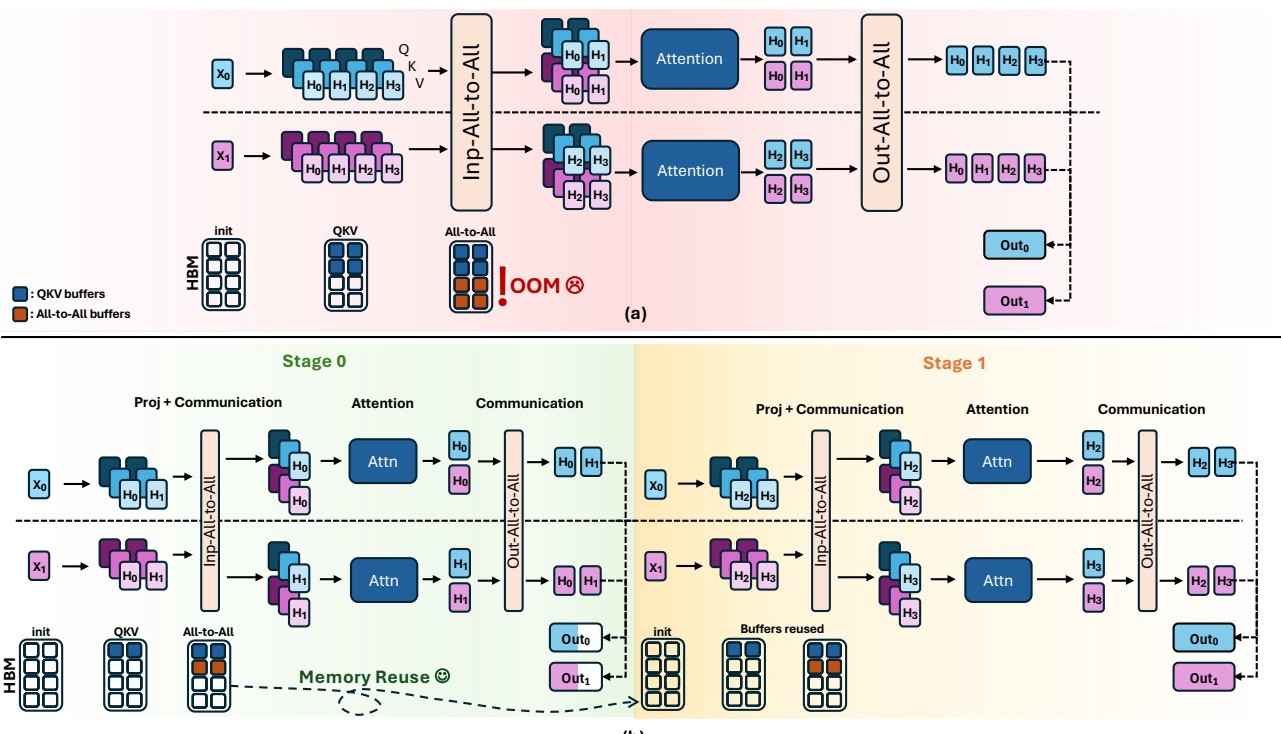

*Figure 3.* Illustration of (a) DeepSpeed-Ulysses and (b) UPipe designs. UPipe processes attention in a headwise *untied* manner, so that in each stage, attention is performed only on a subset of heads. This allows memory reuse across different stages, significantly reducing the peak memory usage due to attention activations. HBM (High-Bandwidth Memory) usage illustrates memory utilization of the intermediate buffers and omits other components for brevity.

### 3.3. Untied Ulysses: UPipe

The biggest memory overhead in DS-Ulysses comes from storing the QKV tensors and corresponding all-to-all buffers *for all heads*. UPipe addresses this by untying the entire attention execution in an end-to-end manner. We propose a headwise chunking scheme for performing attention, which processes only a subset of heads at a time, reducing the peak activation memory. To realize the memory benefits in practice, we also propose a GQA-scheduling technique that processes attention heads out-of-order and prevents redundant communication of key and value heads.

Formally, given a model with attention heads $H$ and number of context parallel devices $C$, UPipe chunks the attention execution into $H/U$ stages, processing $U$ heads per stage. During the forward pass, the execution begins by projecting the input $X$ into $Q_U^0$, $K_U^0$, and $V_U^0$, i.e., the first $U$ heads. This is followed by **inp_all_to_all**, resulting in $U/C$ heads per device. Note that $U$ must be divisible by $C$ to ensure that each device processes an integer number of heads.

In the next stage, we process the next $U$ heads while reusing the memory buffers from the previous stage (i.e., use $Q_U^0$ buffers to store $Q_U^1$ and similarly for other tensors). Thus, the memory usage remains $\mathcal{O}(U)$ throughout the execution.

Let us now walk through the example in Figure 3b with $H = 4$, $U = 2$, and $C = 2$. UPipe performs attention over $H/U = 2$ stages: processing 2 heads per stage.

Consider the execution on **Device 0**: in stage 0, the input $X_0$ is projected into the first two heads $H_0$, $H_1$ of $QKV$. Next, **inp_all_to_all** is performed on $H_0$, $H_1$ so that device 0 has $H_0$ for the entire sequence. Notice that during all-to-all, we only need buffers for 2 heads (as opposed to 4 heads in DS-Ulysses). We then perform attention on $H_0$, generate output for head 0, and perform **out_all_to_all**. In the second stage, $X_0$ is projected into the next two heads $H_2$, $H_3$. At this point, heads $H_0$, $H_1$ are already processed, so we reuse their HBM buffers to store $H_2$, $H_3$. Next, we perform all-to-all and similarly reuse the all-to-all buffers from stage 0. For the final output, we initialize the buffers at the beginning and fill them during execution. This avoids the concatenation of individual output chunks at the end of the attention stage, which otherwise degrades performance.

### 3.4. Memory Savings

For $H$ heads and $C$ devices used for context parallelism, DS-Ulysses has a total memory usage of: $6 \cdot \frac{S}{C} \cdot H \cdot d_{head}$ bytes for the $QKV$ tensors and the same number of bytes for the all-to-all communication buffers, resulting in a total of $12 \cdot \frac{S}{C} \cdot H \cdot d_{head}$ bytes of intermediate tensors.

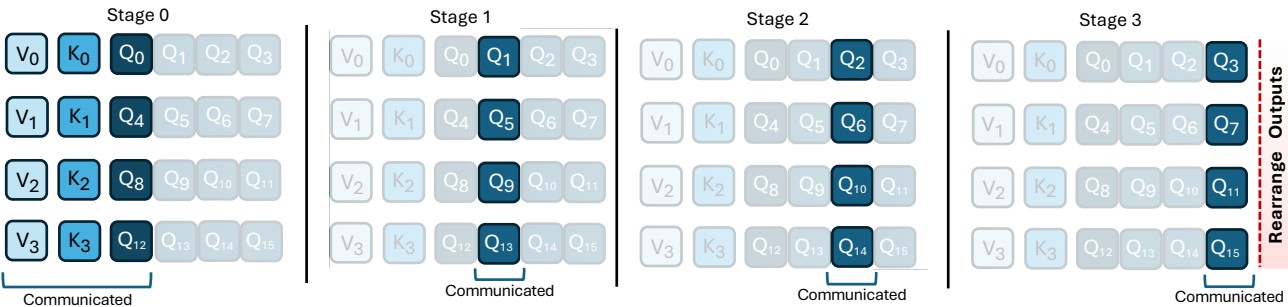

*Figure 4.* Illustration of UPipe's GQA scheduling algorithm. We communicate as many unique key/value heads as possible, along with the corresponding queries in stage 0. In the subsequent stages, we only communicate the next queries of the corresponding groups, reusing the key/value tensors from stage 0 until stage $G$, where $G$ is the group size.

In case of UPipe, we can replace $H$ with $U$, because a single stage processes $U$ heads. Thus, the memory usage is $12 \cdot \frac{S}{C} \cdot U \cdot d_{head}$ bytes. To maximize memory savings, we want $U$ to be as small as possible, and the smallest valid value is $U = C$. In this setting, the maximum memory usage of UPipe during the attention stage becomes $12 \cdot S \cdot d_{head}$. Hence, when $U = C$, the peak activation memory usage of our method is independent of the number of heads.

Taking Qwen3-32B as an example with $H = 64$ and using a single $8\times$H100 node (i.e., $C = 8$), DS-Ulysses requires $96 \cdot S \cdot d_{head}$ bytes of memory. By contrast, UPipe requires $12 \cdot S \cdot d_{head}$ bytes of memory, lowering the memory usage due to intermediate activations for self-attention by **87.5**%.

## 4. Implementation

We use TorchTitan (Liang et al., 2025) as the training framework to integrate the implementation of UPipe due to its flexibility and performance. We use the same framework for Unified Sequence Parallelism (USP) baselines to ensure a fair comparison across different CP techniques.

We integrate the optimized layers inside TorchTitan via a simple drop-in replacement of the existing ones. As the baseline for Ulysses and Ring Attention, we use the hybrid attention implementation from USP, which includes a load-balanced zig-zag Ring Attention implementation. UPipe uses the same code structure as USP, so it can be easily extended for a hybrid setup (Ulysses+Ring). For all-to-all communication, we use the non-QKVPacked variant from USP, which transfers queries, keys, and values sequentially to avoid memory overhead of simultaneous transfers.

For operations independent along the sequence length, such as FFN and Root Mean Squared Normalization (RM-SNorm), we use TiledCompute — a tiling mechanism introduced by ALST (Bekman et al., 2025). We use tiling for RMSNorm, since we found it to be more memory-efficient than using `torch.compile`. Similarly to the authors of ALST, we use a square tile of size $[d_{model} \times d_{model}]$. We use `FusedLinearCrossEntropyLoss` from Liger-

Kernel (Hsu et al., 2025) for memory-efficient loss computation. This kernel fuses the final linear projection with the cross-entropy loss, computing logits and loss in a tiled manner to avoid materializing the full `fp32` logits tensor in memory. We also find that Rotary Positional Encoding (Su et al., 2023) also incurs a memory overhead due to `fp32` casting. Hence, we use the fused RoPE implementation from the Flash Attention API, which performs in-place operations to avoid large memory spikes.

### 4.1. Grouped-Query Attention Scheduling

Because of its memory benefits, GQA is an important architectural component of modern Transformer models (Grattafiori et al., 2024; Jiang et al., 2023; Yang et al., 2025). To retain this advantage, we make our design GQA-compatible. This requires rearranging the query tensors to reuse the KV tensors communicated in the previous stage.

Assuming $U = C$, recall that UPipe processes $C$ heads per stage. For a GQA model with group size $G$, only $C/G$ of the first $C$ heads are unique. For example, consider $C = 4$ and $G = 4$. With naive processing, stage 0 would process query heads $Q_0, Q_1, Q_2, Q_3$ with the corresponding key heads $K_0, K_0, K_0, K_0$. Since each device transfers $C - 1$ query, key, and value heads per stage, the total communication volume across $H/C$ stages is $\mathcal{O}(3 \cdot \frac{H}{C} \cdot (C - 1)) = \mathcal{O}(3H)$.

As shown in Figure 4, we could communicate $K_0, K_1, K_2, K_3$ (i.e., all the unique key tensors) in stage 0. We would then need to send the corresponding query tensors (i.e., $Q_0, Q_4, Q_8, Q_{12}$). However, in the next stage, we do not need to communicate any key (or value) tensors. We can now choose the next query tensors from the groups (i.e., $Q_1, Q_5, Q_9, Q_{13}$), because the corresponding key (and value) tensors were already communicated in stage 0. In this case, for every $G$ stages, we communicate $C - 1$ query, key, and value heads in the first stage, and only $C - 1$ query heads in the following stages. Thus, the total communication volume is $\mathcal{O}\left((3 + G - 1) \cdot \frac{H}{C \cdot G} \cdot (C - 1)\right) = \mathcal{O}\left(\frac{2H}{G} + H\right)$, which is always less than the naive processing, because $G > 1$.

# 5. Experiments

We now present the experimental results using UPipe, comparing it with multiple state-of-the-art baselines, as well as standard context parallelism approaches. We run the experiments on two models across multiple context lengths, comparing the maximum context length supported by different techniques and their throughput in each setup. Unless stated otherwise, for all experiments with UPipe, we match the chunk size with the number of context-parallel devices ($U = C$) to reach the highest memory efficiency.

## 5.1. Setup

We run our experiments on $8\times$ NVIDIA H100 nodes with 80GiB of on-chip DRAM per GPU, connected via 4th generation NVLink with 900GBps bidirectional bandwidth for intra-node communication and 400 Gbps Infiniband networking for inter-node communication. Each node has a Intel Xeon Platinum 8480+ CPU with 1.9TiB RAM.

We use Llama 3-8B (Grattafiori et al., 2024) and Qwen3-32B (Yang et al., 2025) models in our experiments. Llama 3-8B has $H = 32$ query heads (and 8 key-value heads), allowing UPipe to process attention in $H/C = 4$ stages. Qwen3-32B has 64 query heads (and 8 key-value heads), so UPipe processes it in 8 stages. Note that we use $C = 8$ here, because we always restrict the Ulysses context parallelism degree to 8 and use the remaining mesh for Ring Attention in a hybrid style, as discussed in Section 5.2.

Our experiments use TorchTitan and Flash Attention-3 (FA3, Shah et al., 2024). We use context parallelism over $8\times$H100 nodes for our training. We use PyTorch's FSDP2 (natively supported by TorchTitan) to distribute parameters, gradients, and optimizer states across all GPUs.

To manage activation memory efficiently and maintain consistency with FPDT, we use full activation checkpointing with CPU offloading. For all sequence lengths except 5M, we allow the offloaded activations to reside in the non-swappable CPU RAM by setting `pin_memory` to True. For the 5M context, we set this to False due to the CPU RAM constraints. For better memory management, we set `PYTORCH_CUDA_ALLOC_CONF =expandable_segments:True`, similarly to ALST.

## 5.2. Baselines

We compare the performance and memory efficiency of UPipe against multiple baselines. Fully Pipelined Distributed Transformer (FPDT, Yao et al., 2025) is a long-context training method that processes attention by chunking along the sequence length dimension and using online softmax to perform full attention. It mitigates the memory requirements by asynchronously offloading chunks to the CPU, keeping only the necessary chunks on the GPU, and using a double buffer mechanism to overlap communication with computation. It can support a maximum sequence of 4M tokens when training Llama 3-8B on a single $8\times$H100. The original implementation of FPDT does not natively support Flash Attention 3 or SwiGLU activations, which is why we patched it for a fair comparison with UPipe.

For standard Ulysses and Ring Attention baselines, we use the USP implementation, as it provides an easy-to-use interface for modifications. Additionally, we also compare with the native Ring Attention implementation from PyTorch. Our experiments omit ALST, because our modified version of USP-Ulysses that uses DS-Ulysses, offloaded activation checkpointing, and tiling for MLP / CELoss integrates all improvements from the ALST design. However, unlike ALST, we do not offload optimizer states to the CPU to avoid throughput degradation.

For multi-node experiments, we use a similar setup as USP-Hybrid with *8-ulysses-2-ring*, which denotes using Ulysses over 8 GPUs within the node and Ring Attention across 2 nodes. This is a common setup (e.g., Team Wan et al., 2025) to allow faster all-to-all communication for Ulysses within the node, and slower ring communication across nodes. Note that FPDT does not support hybrid context parallelism, so we use the standard *16-ulysses-1-ring* setup.

We also compare against pure Ring Attention baselines: USP-Ring and a native PyTorch implementation. Both of them use zig-zag load balancing, ensuring fair work distribution across all ranks participating in context parallelism.

## 5.3. Performance

### 5.3.1. SINGLE-NODE TRAINING

**Llama 3-8B:** Table 3 (top) reports single-node throughput for Llama 3-8B training across increasing sequence lengths. Ulysses has high throughput due to a single all-to-all call per attention, but its full-head $QKV$ and communication buffers make activation memory the primary limiter at multi-million token contexts. FPDT can push context length further, but it pays a substantial throughput penalty from frequent CPU–GPU transfers due to fine-grained CPU offloading.

UPipe bridges this gap: at shorter sequences, it is slightly slower than Ulysses due to additional stage launches, but this overhead is amortized with longer contexts (as shown in Table 9). At longer sequences ($\geq$2M), UPipe matches Ulysses throughput while using much lower GPU memory, enabling training with 5M tokens on a single $8\times$H100 node. In effect, UPipe preserves the throughput of Ulysses while having the memory efficiency closer to offloading-based methods, improving the maximum single-node context length over FPDT (4M tokens) by **25%**.

*Table 3.* **Throughput comparison** (tokens/second/GPU) for **Llama 3-8B** (8×H100s) and **Qwen3-32B** (16×H100s) across varying sequence lengths. On a single 8×H100 node, our method scales Llama 3-8B to a 5M-token sequence length, reaching a 25% improvement over the previous state-of-the-art. **OOM**: Out of Memory. **Note**: FPDT execution fails at lengths > 4M.

| | Method | 128K | 256K | 512K | 1M | 2M | 3M | 4M | 5M |
|---|---|---|---|---|---|---|---|---|---|
| **Llama 3-8B** | Native PyTorch | 1373.87 | 845.99 | 474.30 | 249.85 | OOM | – | – | – |
| | Ring | 2064.90 | 1387.67 | 841.05 | 458.51 | 237.99 | 159.96 | OOM | – |
| | Ulysses | **2320.47** | **1503.80** | **878.63** | **475.33** | 246.05 | 162.41 | OOM | – |
| | FPDT | 1171.68 | 884.75 | 621.20 | 382.42 | 219.53 | 153.48 | 119.76 | – |
| | UPipe | 2281.05 | 1487.29 | 867.17 | 472.53 | **246.07** | **166.32** | **125.56** | **98.25** |
| **Qwen3-32B** | Native PyTorch | 127.03 | 112.20 | 91.39 | OOM | – | – | – | – |
| | Ring | 418.39 | 308.88 | 194.44 | 110.27 | 58.45 | OOM | – | – |
| | Ulysses | **545.29** | **370.70** | **217.04** | **117.02** | **59.98** | OOM | – | – |
| | FPDT | 286.40 | 217.85 | 151.91 | 95.88 | 55.41 | 38.86 | 27.66 | – |
| | UPipe | 483.29 | 339.56 | 204.46 | 113.26 | **59.56** | **40.42** | **29.97** | OOM |

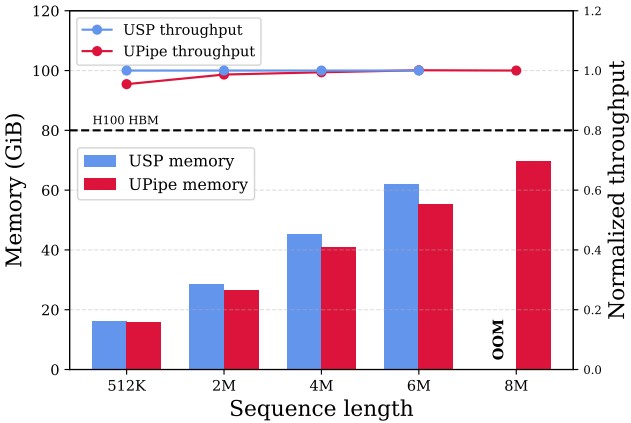

*Figure 5.* **Llama 3-8B:** Comparison of peak GPU memory usage and throughput (normalized w.r.t. USP-Hybrid) between UPipe and USP-Hybrid at different sequence lengths on 16×H100s.

### 5.3.2. MULTI-NODE TRAINING

**Llama 3-8B:** Figure 5 shows the comparison of UPipe and USP-Hybrid on a 16×H100 setup. UPipe is more memory efficient than USP-Hybrid at all sequence lengths from 512K to 6M tokens, and supports context lengths up to 8M tokens, improving upon USP-Hybrid by **33%**. The throughput of UPipe is comparable to USP-Hybrid at all sequence lengths, highlighting UPipe's runtime efficiency.

**Qwen3-32B:** Table 3 (bottom) summarizes the throughput comparison of different methods when training Qwen3-32B on a 16×H100 setup. UPipe outperforms all other methods for sequences ≥ 2M. Notably, UPipe always outperforms FPDT across all sequence lengths in terms of throughput. In terms of maximum sequence length, our method can support 4M token sequences, 2× more than Ulysses (2M tokens), while delivering **8.3%** better performance than FPDT.

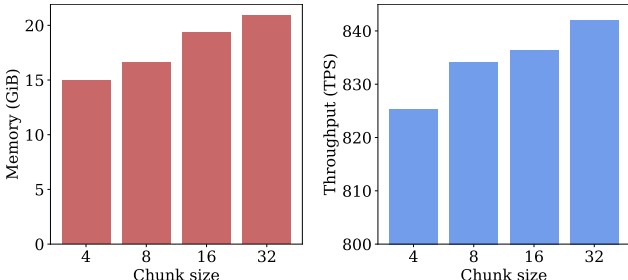

*Figure 6.* Ablation analysis of UPipe's head-chunk size $U$ when training Llama 3-8B with a context size of 512K on $C = 4$ GPUs.

Additionally, we also report the maximum allocated memory in Table 8 of Appendix B. UPipe provides the best memory efficiency compared to other methods except FPDT. However, FPDT suffers from performance degradation due to frequent CPU transfers. Nevertheless, our method should be composable with FPDT due to orthogonal chunking dimensions, allowing benefits from both the methods.

### 5.4. Ablation on the head chunk size

For the above experiments, we aimed to show the maximum memory efficiency of UPipe, so we chose $U = C$. Next, we present an ablation on $U$, discussing the associated tradeoffs between memory and throughput. Specifically, we analyze the memory-runtime tradeoff associated with the number of chunks and the size of each chunk. We use Llama 3-8B on 4×H100 GPUs with a context size of 512K tokens.

As shown in Figure 6, increasing $U$ corresponds to more heads processed per stage, resulting in higher memory usage and lower runtime. Conversely, when $U = C$, UPipe provides maximum memory benefits at the cost of slight performance degradation due to kernel launch overhead. However, at longer context lengths, this overhead is amortized by the increased length, as shown in Table 9 of Appendix C.

## 6. Reinvesting Memory Savings

UPipe consumes significantly less memory than Ulysses. However, for context lengths in the 128K–1M range, UPipe falls slightly behind Ulysses in terms of performance. In this section, we demonstrate the potential performance gains that could be obtained by reinvesting UPipe's memory savings.

### 6.1. Strided Activation Offloading

As mentioned in Section 5.1, we employ full activation checkpointing with CPU offloading. Since UPipe provides more GPU memory headroom, we explore **strided activation offloading**, which skips offloading the activations of every $K^{th}$ layer. This eliminates the CPU–GPU transfer overhead for every $K^{th}$ layer, resulting in an overall performance improvement.

*Table 4.* Comparison of throughput (TPS) and memory usage (GiB) across Ulysses, UPipe, and UPipe with **SAO** (Strided Activation Offloading). **SAO** narrows the performance gap between Ulysses and UPipe due to less frequent activation offloading to CPU.

|     | Method | 128K | 256K | 512K | 1M |
|-----|--------|------|------|------|-----|
| TPS | Ulysses | 2417.79 | **1516.51** | **880.27** | **476.45** |
|     | UPipe | 2377.98 | 1515.59 | 878.73 | 470.34 |
|     | UPipe-SAO | **2445.10** | 1512.69 | 872.57 | 471.07 |
| GiB | Ulysses | 62.94 | 63.01 | 63.15 | 63.43 |
|     | UPipe | 53.24 | 53.31 | 53.45 | 53.73 |
|     | UPipe-SAO | 62.39 | 62.47 | 62.61 | 62.89 |

We conduct strided activation offloading experiments on Llama 3-8B using a single 8×H100 node. We use a batch size > 1 such that the total number of tokens is always 3M. We use $K = 7$, i.e., skip CPU activation offloading every 7 layers. Table 4 compares throughput and memory usage across Ulysses, UPipe and UPipe-SAO. For the context length of 128K, SAO significantly improves performance over both Ulysses and UPipe. This is because at shorter sequence lengths, the CPU–GPU transfers contribute significantly to the overall runtime. Skipping CPU offloading for every $7^{th}$ layer results in **2.8%** improvement in performance for 128K. For longer sequence lengths, we see minimal change, because the quadratic scaling of attention computation time dominates the overall runtime.

### 6.2. Selective Activation Checkpointing

Selective activation checkpointing (SAC, Korthikanti et al., 2022) retains the output activations of attention to prevent costly recomputation during the backward pass. The forward pass of UPipe is implemented as a custom PyTorch operator, enabling off-the-shelf support for SAC. As shown in Table 5, enabling SAC further improves the performance of UPipe, while keeping its memory usage lower than that of DeepSpeed-Ulysses (both with and without SAC).

*Table 5.* Throughput (in tokens/second) and memory usage (GiB) of Ulysses and UPipe with and without Selective Activation Checkpointing at a sequence length of 512K for Llama 3-8B.

| Method | Throughput | Memory |
|--------|-----------|--------|
| Ulysses | 864.16 | 27.34 |
| Ulysses+SAC | **1066.01** | 27.85 |
| UPipe | 850.97 | **25.24** |
| UPipe+SAC | 1056.77 | 25.75 |

### 6.3. Batch size ablation

Due to the lower memory usage of UPipe, it is also possible to reuse the freed up memory by increasing the per-device batch size. Hence, we analyze the training throughput by varying the batch size for Llama3-8B training on a single 8×H100 node at 128K context length. We turn off CPU offloading of activations to avoid performance variability due to the PCIe bus traffic. As shown in Table 6, the throughput increases monotonically for batch sizes smaller than 8. However, at the batch size of 10, the speed of Ulysses degrades due to memory pressure triggering frequent CUDA allocation retries. UPipe avoids this through better memory management, outperforming Ulysses at the largest batch size while operating at lower memory across all batch sizes.

*Table 6.* Comparison of throughput (TPS) and memory usage (GiB) for Ulysses and UPipe across different batch sizes.

|     | Batch size | 1 | 2 | 4 | 8 | 10 |
|-----|-----------|------|------|------|------|------|
| TPS | Ulysses | **2458.28** | **2485.78** | **2543.43** | **2564.38** | 2464.10 |
|     | UPipe | 2405.30 | 2432.11 | 2498.80 | 2518.56 | **2526.77** |
| GiB | Ulysses | 22.53 | 27.25 | 38.13 | 59.99 | 71.13 |
|     | UPipe | **22.53** | **27.03** | **36.56** | **56.79** | **67.11** |

## 7. Conclusion

In this paper, we introduced UPipe, a context parallelism method that reduces the attention activation memory usage by chunking attention at the head level. Our experiments demonstrate that these memory gains allow further context scaling without sacrificing throughput. On Llama 3-8B, UPipe reaches **5M** tokens on a single 8×H100 node (**25%** beyond FPDT), and scales to **8M** tokens on 16 H100 GPUs, while keeping throughput comparable to common baselines. For larger models, UPipe reduces the intermediate tensor memory usage of attention by up to 87.5%, helping avoid allocation retries that otherwise degrade training performance.

Overall, UPipe provides a practical path to pushing the context length frontier: it is simple, composable with existing methods, and offers substantial memory usage reductions while preserving the training speed. While the study of UPipe could be extended (see Appendix E for an overview of limitations), we expect this method be a useful building block for future systems, especially as new tasks and modalities continue to demand larger sequence lengths.

## Acknowledgements

We thank the anonymous reviewers for their thoughtful and constructive feedback, which substantially improved this work. We are also grateful to the developers and maintainers of TorchTitan (Liang et al., 2025) and USP (Fang & Zhao, 2024), whose open-source projects provided a strong basis for the implementation and evaluation of UPipe. More broadly, we thank the open-source and research community, whose tooling and prior results made this work possible.

## Impact Statement

This work aims to advance the field of Machine Learning by improving the memory efficiency of training transformer based models on long context inputs. Our work extends the maximum context length that can fit on a given hardware, and thus is broadly applicable to a variety of long-context settings for Transformer models, but it is not designed for any particular use case.

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

## Supplementary Material

## A. Per-stage memory accounting for the attention block

In Tables 2 and 7 we list the peak activation memory of the attention block under GQA at each stage, for DeepSpeed-Ulysses with and without activation offloading, FPDT, and UPipe. This section walks through how the entries for the Ulysses and UPipe rows are obtained.

*Table 7.* Peak activation memory during the backward pass of the attention stage under GQA, expressed in units of one local hidden-state shard ($\frac{S}{C} \cdot d_{model}$ `bf16` elements, i.e., $2 \cdot \frac{S}{C} \cdot d_{model}$ bytes). $\pi$ represents the number of chunks in FPDT, $\nu$ represents the number of chunks in UPipe.

| Method | Before Bwd Attn | During out_all_to_all | During Bwd Attn Kernel | During inp_all_to_all |
|---|---|---|---|---|
| Ulysses | $L+2$ | $L+3$ | $L+2\gamma+2$ | $L+\gamma+1$ |
| Ulysses + offloading | 3 | 4 | $2\gamma+3$ | $\gamma+2$ |
| FPDT | $\frac{1}{\pi}$ | $\frac{3}{\pi}$ | $\frac{2\gamma+4}{\pi}$ | $\frac{\gamma+2}{\pi}$ |
| Untied Ulysses | 3 | $3+\frac{1}{\nu}$ | $\max\left(3+\frac{2\gamma}{\nu},\, 2+\frac{2\gamma}{\nu}+\frac{2(\nu-1)}{\nu}\right)$ | $2+\frac{\gamma}{\nu}+\frac{2(\nu-1)}{\nu}$ |

**DeepSpeed-Ulysses (forward).** On entry to the attention block, only the saved layer input $X$ is live on GPU. During inp_all_to_all, the projected $Q, K, V$ ($\gamma$ units) and one head-distributed destination buffer (1 unit) are alive in addition to $X$ (1 unit). Because the non-QKVPacked all-to-all communicates $Q, K, V$ sequentially, only one destination buffer is in flight at a time, and the peak (for $G > 1$) occurs during $Q$'s exchange: $1\,(X) + \gamma\,(Q, K, V) + 1\,(Q') = \gamma + 2$. During the attention kernel, the live set is $\{X, Q', K', V', O'\}$, again summing to $\gamma + 2$. During out_all_to_all, the head-distributed buffers are released and only $\{X, O', O\}$ remain (3). Without activation offloading, the inputs of the preceding $L - 1$ layers are also held on GPU, which we account for by the leading $L$ term of the Ulysses row.

**DeepSpeed-Ulysses (backward).** The block begins with three sequence-sharded 1-unit tensors: $X$ (the layer input, saved by the framework for the QKV-projection weight gradient), $O$ (the saved attention block output, kept in sequence-sharded layout because it is the per-token input to proj_o and thus required for the proj_o weight gradient), and the upstream gradient $\delta_O$ on $O$, summing to 3. The FlashAttention backward kernel, however, needs both quantities in head-distributed form, so two all-to-alls are issued during out_all_to_all: one on $O$ (producing $O'$) and one on $\delta_O$ (producing $\delta'_O$). Running these sequentially with slot reuse—the buffer of the first all-to-all's input is freed after the call and reused as the destination of the second—caps the peak at 4: $X$, one in-flight source, its destination, and the still-alive second source. The FlashAttention backward kernel then requires $\{Q', K', V', O', \delta'_O, \delta Q', \delta K', \delta V'\}$ simultaneously (a $(2\gamma + 2)$-unit footprint) on top of $X$ (the seq-sharded $O$ and $\delta_O$ have been consumed in the previous step), yielding $2\gamma + 3$. The inverse inp_all_to_all mirrors the forward and peaks at $\gamma + 2$.

**Untied Ulysses (forward).** UPipe executes the attention block in $\nu = H/U$ stages, each processing $U$ heads. A full-sized output buffer final_out (1 unit) is pre-allocated at the start of the block. Because final_out is initially empty, slot $i$ of size $1/\nu$ can be transiently repurposed during stage $i$ to hold the head-distributed query $Q'^i$, eliminating a separate destination buffer for that all-to-all. At inp_all_to_all of stage $i$, the live tensors are $X$, final_out (with slot $i$ doubling as $Q'^i$'s buffer), and the chunked $Q^i, K^i, V^i$ ($\gamma/\nu$), giving $2 + \gamma/\nu$. At the attention kernel, $Q'^i$ still resides in slot $i$ of final_out, $K'^i, V'^i$ ($2/(G\nu)$) and the head-distributed output $O'^i$ ($1/\nu$) coexist separately, summing again to $2 + \gamma/\nu$. At out_all_to_all, $O'^i$ is gathered into slot $i$ of final_out; the only transient cost is one in-flight chunk, yielding $2 + 1/\nu$. Across the whole stage, the intermediate buffers (chunked $Q, K, V$, all-to-all in/out, attention output) are $\nu$ times smaller than their Ulysses counterparts, at the price of two persistent 1-unit tensors ($X$ and final_out) instead of one.

**Untied Ulysses (backward).** The saved final_out and its upstream gradient $\delta_{\text{final\_out}}$ each remain alive throughout the $\nu$-stage loop, contributing 2 units on top of $X$ on entry to the block. Each stage consumes one slot of final_out and $\delta_{\text{final\_out}}$ via per-chunk all-to-alls; the freed slot is reused as the destination for the corresponding chunk's intermediate buffer, exactly as in the forward pass. A new full-sized accumulator $\delta_X$ (1 unit) for the gradient w.r.t. the layer input is allocated at the end of stage 0's inverse all-to-all and updated by every subsequent stage. $\delta_X$ cannot share storage with a

freed slot of `final_out` or $\delta_{\texttt{final\_out}}$: each per-stage matmul $\delta Q^i \cdot W_q^{(i)}$ contributes to *all* $d_{model}$ components of $\delta_X$ at once, so $\delta_X$ must be a contiguous full-sized buffer that exists from stage 0 onward, while slots are freed only one at a time.

The `inp_all_to_all` column reports the peak at the end of stage 0, immediately after the inverse all-to-all returns and $\delta_X$ is allocated for the first time. At that moment, the live tensors are $X$ (1), $\delta_X$ (1), the remaining slots of `final_out` and $\delta_{\texttt{final\_out}}$ after slot 0 has been consumed (each $(\nu - 1)/\nu$), and the Q-side gradient $\delta Q^0$ together with the per-group key/value accumulators $\delta K^0, \delta V^0$ ($\gamma/\nu$ combined), giving

$$2 + \gamma/\nu + 2(\nu - 1)/\nu.$$

At $\nu = 1$ this collapses to $\gamma + 2$, matching Ulysses with activation offloading.

The FlashAttention backward kernel column reports the maximum of two per-stage formulas to capture the global backward peak across all stages. The first argument, $3 + 2\gamma/\nu$, is the stage-0 peak: $X + \texttt{final\_out} + \delta_{\texttt{final\_out}} + (\delta Q'^0, \delta K'^0, \delta V'^0) + (Q'^0, K'^0, V'^0)$ (no $\delta_X$ yet). The second argument, $2 + 2\gamma/\nu + 2(\nu - 1)/\nu$, is the stage-1 peak with $\delta_X$ live and slot 0 of `final_out`, $\delta_{\texttt{final\_out}}$ already consumed: $X + \delta_X + (\nu - 1)/\nu\,(\texttt{final\_out} + \delta_{\texttt{final\_out}}) + (\delta Q'^1, \delta K'^1, \delta V'^1) + (Q'^1, K'^1, V'^1)$. At $\nu = 1$ stage 1 does not exist; the cell equals $2\gamma + 3$ (corresponding to stage 0), matching Ulysses with offloading. At $\nu = 2$ the two arguments are equal; for $\nu \geq 3$ the stage 1 form is strictly larger and dictates the cell value.

The `out_all_to_all` column is reported w.r.t stage 0 equating to $3 + 1/\nu$, with $\nu = 1$ corresponding to Ulysses with offloading; for $\nu \geq 2$ its per-stage peak is also 1 unit larger when $\delta_X$ is live, but this is not the global peak. The *before backward attention* column remains constant (at 3) because $\delta_X$ has not yet been allocated.

## B. Memory usage comparison

*Table 8.* **Memory comparison** (GiB) for **Llama 3-8B** (8×H100s) and **Qwen3-32B** (16×H100s) across varying sequence lengths. **OOM** denotes Out of Memory. FPDT execution fails at lengths $> 4M$. The lowest memory usage at a given sequence length is in bold.

| Model | Method | 128K | 256K | 512K | 1M | 2M | 3M | 4M | 5M |
|---|---|---|---|---|---|---|---|---|---|
| | Native PyTorch | 25.32 | 31.40 | 43.55 | 67.86 | OOM | – | – | – |
| | Ring | 21.32 | 23.40 | 27.58 | 35.86 | 52.49 | 69.11 | OOM | – |
| Llama 3-8B | Ulysses | 21.26 | 23.02 | 26.80 | 34.35 | 49.49 | 64.55 | OOM | – |
| | FPDT | 21.73 | 22.50 | **24.03** | 27.09 | **35.17** | **43.35** | **51.42** | – |
| | UPipe | **21.10** | **22.30** | 24.70 | 29.90 | 40.50 | 51.10 | 61.70 | **72.30** |
| | Native PyTorch | 45.81 | 53.69 | 69.47 | OOM | – | – | – | – |
| | Ring | 40.14 | 41.16 | 44.22 | 50.51 | 63.11 | OOM | – | – |
| Qwen3-32B | Ulysses | 40.13 | 41.16 | 44.10 | 50.27 | 62.60 | OOM | – | – |
| | FPDT | **38.94** | **39.47** | **40.54** | **42.66** | **46.91** | **52.27** | **57.77** | – |
| | UPipe | 39.98 | 40.84 | 42.72 | 46.84 | 55.65 | 64.47 | 73.28 | OOM |

Table 8 shows the memory usage comparison of various context parallelism schemes on Llama 3-8B and Qwen3-32B models across different context lengths. FPDT exhibits the best memory usage, but performs poorer due to the CPU overhead. UPipe has better memory efficiency than all other methods, while also matching the throughput of Ulysses. Note that while FPDT reports lower allocated memory, it is unable to run with context lengths greater than 4M.

## C. Runtime comparison between DeepSpeed-Ulysses and UPipe

*Table 9.* Runtime comparison (in seconds) for Llama 3-8B on 8×H100 between DeepSpeed-Ulysses and UPipe across sequence lengths. **FA3-Fwd/Bwd:** Total Flash Attention-3 forward and backward kernel time. **Total** refers to the total time for a single training step.

| Method | Stage | 128K | 256K | 512K | 1M | 2M | 3M |
|---|---|---|---|---|---|---|---|
| Ulysses | All-to-All | 0.40 | 0.90 | 1.68 | 4.93 | 16.30 | 42.21 |
| | FA3-Fwd | 1.58 | 6.35 | 25.71 | 103.49 | 421.67 | 995.92 |
| | FA3-Bwd | 2.40 | 9.13 | 36.74 | 146.86 | 588.73 | 1324.71 |
| | Other | 3.03 | 5.33 | 10.08 | 19.78 | 41.30 | 56.31 |
| | **Total** | **7.40** | **21.72** | **74.21** | **275.06** | **1068.00** | **2419.14** |
| UPipe | All-to-All | 0.46 | 1.10 | 2.43 | 5.52 | 17.12 | 34.34 |
| | FA3-Fwd | 1.51 | 6.38 | 25.93 | 103.92 | 417.55 | 940.62 |
| | FA3-Bwd | 2.41 | 9.25 | 36.99 | 147.37 | 590.79 | 1330.76 |
| | Other | 2.82 | 5.23 | 10.10 | 19.58 | 37.76 | 55.52 |
| | **Total** | **7.20** | **21.96** | **75.45** | **276.39** | **1063.23** | **2361.24** |

Table 9 shows the runtime comparison for Llama 3-8B on a single 8×H100 node between DS-Ulysses and UPipe. It shows the breakdown of the runtime of a single training step into major components: Flash Attention-3 forward time, Flash Attention-3 backward time, and All-to-All communication time. Note that UPipe has a higher runtime at shorter sequence lengths due to multiple kernel launches. At longer sequence lengths, this disadvantage is amortized, because every kernel executes more computational operations and starts to operate in the compute-bound regime. Note that the table above reports the time per training step, so it exhibits greater variance than the throughput numbers in Table 3 (especially at shorter sequence lengths).

## D. Training Loss Convergence

*Table 10.* Loss Convergence: Comparison of training loss between UPipe and Ulysses when training a Llama 3-8B model on C4 dataset with sequence length of 128K tokens over 1000 steps.

| Method | 100 | 200 | 300 | 400 | 500 | 600 | 700 | 800 | 900 | 1000 |
|---|---|---|---|---|---|---|---|---|---|---|
| Ulysses | 6.83 | 6.58 | 5.98 | 5.62 | 5.22 | 4.88 | 4.94 | 5.11 | 4.68 | 4.47 |
| UPipe | 6.85 | 6.54 | 5.98 | 5.62 | 5.22 | 4.90 | 4.95 | 5.11 | 4.69 | 4.47 |
| %Diff | 0.29% | -0.61% | 0.00% | 0.00% | 0.00% | 0.41% | 0.20% | 0.00% | 0.21% | 0.00% |

UPipe performs the exact same computation during the attention stage as Ulysses. However, our proposed GQA scheduling strategy changes the order in which different attention heads are processed. This could lead to minor numerical differences due to the non-associative nature of floating point operations. In this section, we verify that UPipe does not affect the training loss convergence by comparing the training dynamics with DeepSpeed-Ulysses on Llama 3-8B.

We train this model on a single 8×H100 node with a sequence length of 128K tokens, a batch size of 1, and a learning rate of $10^{-5}$. We use the C4 dataset (Raffel et al., 2023) for training the model on 1000 steps, and report the loss after every 100 steps in Table 10.

## E. Scope and Limitations

UPipe applies broadly to Transformer-based models (LLMs, Vision Transformers, Diffusion Transformers) regardless of the modality, as the only component it modifies is multi-head self-attention. Although its throughput at shorter contexts is lower than DeepSpeed-Ulysses, better memory efficiency enables larger batch sizes and Selective Activation Checkpointing (Appendix 6.2). Using an all-to-all implementation that does not involve Streaming Multiprocessors (also known as "SM-free communication") could further improve throughput by overlapping communication with attention computation.

