# OpenReview forum: "Untied Ulysses: Memory-Efficient Context Parallelism via Headwise Chunking"
_ICML.cc/2026/Conference — ICML 2026 regular_

### Official Review · Reviewer_1W6p · 2026-03-02

**Soundness:** 4
**Presentation:** 3
**Significance:** 4
**Originality:** 4
**Overall Recommendation:** 5
**Confidence:** 3

**Summary:**

This manuscript starts with the premise that memory consumption is a limiting factor for processing transformer-based computational graphs in multi-GPUs. The bottleneck is the memory requirement for the all-to-all communication in DeepSpeed-Ulysses, the SOTA method for sharding QKV decomposition and attention computations for multiple GPUs. The paper proposes a new way to process large contexts in parallel with multiple GPUs. In this new solution, the number of heads computed at a time is reduced to ensure that it does not exceed the capacity of a single GPU. Because this reduction saturates the computational capacity of each single GPU, the advantage is that the memory usage is reduced without significantly affecting performance.

**Compliance With Llm Reviewing Policy:**

Affirmed.

**Final Justification:**

I have no further comments. I maintain my previous evaluation of the paper.

**Key Questions For Authors:**

Does it make sense to consider cases U is not divisible by C? If yes, how the computation of the remainder chunks would affect the results in the paper?

**Limitations:**

It would be nice to have a section that discusses the boundaries of applicability of the solution in the views of the author.

The societal impact is obvious because the proposed solution enables the processing of larger contexts with the same number of GPUs, thus reducing the pressure for deploying more devices.

**Strengths And Weaknesses:**

Strengths:
- It is a simple idea that is effective and solves a real problem.
- The paper is well organized and the examples and figures help to understand the concept.
- It is practical: it enables the processing of larger contexts with the same number of GPUs -- the SOTA solution runs out of memory

Weakness:
- At places the manuscript assumes too much prior knowledge and reads as an insider note to others that know the minutiae of attention-based system, rather than a scientific paper for a broader audience.


Nitpick:

Make sure that every acronym is spelled out the first time that it appears. For example GQA architecture in introduction (it is only spelled out in Section 2.3), RMSNorm in Section 2.3

---

> ### Author Rebuttal · Authors · 2026-03-31
>
> We thank the reviewer for their appreciation of our work and for providing thoughtful comments. We have addressed the points raised in the review below:
>
> **Q-1: U divisibility by C**\
> R-1: Ideally, U must be divisible by C. If not, some GPUs will process remainder chunks (as pointed out by the reviewer), causing a straggler effect due to non-uniform work distribution. In our paper, we assume uniform work distribution to avoid this. Note that Ulysses already requires the number of heads H to be divisible by C, and UPipe's constraint naturally follows. Hence, for use cases that support Ulysses, UPipe works out of the box.
>
> On a side note, this is an interesting avenue to explore, particularly when computation per attention head varies. For example, during LLM inference, HeadKV [1] suggests independent KV cache pruning per head, leading to unequal computation across heads. In such scenarios, the divisibility constraint could be relaxed to allow asymmetric work distribution, assigning remainder chunks to GPUs with lower load. A detailed analysis is beyond the scope of our paper, and we leave it as future work.
>
> [1] HeadKV: https://arxiv.org/pdf/2410.19258
>
> **Q-2: Addressing a broader audience**\
> R-2: We have identified specific exposition improvements and will make the following concrete revisions:
>
> **a. Background on attention variants:** We will add a self-contained primer covering: (i) the multi-headed attention mechanism (Q, K, V formulation), (ii) a brief introduction to Grouped-Query Attention (GQA), and (iii) how context parallelism schemes (Ring Attention, DeepSpeed-Ulysses) affect the attention computation in a distributed training setup.
>
> **b. Acronym audit:** We have audited the manuscript and identified acronyms used before being defined (e.g., GQA in Section 1, RMSNorm in Section 2.3). We will spell out each at its first occurrence.
>
> **Q-3: Discussion on boundaries of applicability**\
> R-3: We will add a "Scope and Limitations" section discussing UPipe's constraints and beneficial regimes of operation. Regarding applicability, UPipe is broadly applicable to all Transformer based models that use multi-headed attention to train on very long sequences (such as video models, code assistants etc.).

---

> > ### Author Rebuttal · Reviewer_1W6p · 2026-04-03
> >
> > I have no further comments.
> > I maintain my previous evaluation of the paper.

---

### Official Review · Reviewer_c8BF · 2026-03-09

**Soundness:** 3
**Presentation:** 3
**Significance:** 2
**Originality:** 2
**Overall Recommendation:** 4
**Confidence:** 4

**Summary:**

This paper proposes UPipe: headwise chunking for context parallelism atop DeepSpeed-Ulysses. Splits attention into H/U stages, reusing QKV and all-to-all buffers across stages. Reduces intermediate memory from O(H) to O(U). Claims 82.5% reduction for 70B. On 8×H100 supports 5M tokens (vs FPDT 4M); on 16×H100 supports 9M tokens.

**Compliance With Llm Reviewing Policy:**

Affirmed.

**Final Justification:**

My final recommendation is Weak Accept (4).

The weaknesses are largely resolved. The remaining concern is practical: UPipe reduces memory footprint with only slight throughput overhead, but the experiments do not fully demonstrate the expected "trade-off" payoff in system design. Specifically, at shorter sequence lengths (the most common regime), one would expect the freed memory to be reinvested — via larger batch sizes, reduced activation checkpointing, or other parallelism strategies — to achieve net throughput gains. While the authors provide some evidence of this (e.g., Table 5), the benefit is not consistent across configurations. I encourage the authors to more systematically characterize this memory-throughput tradeoff in future revisions.

**Key Questions For Authors:**

* Numerical equivalence: The paper claims UPipe is a drop-in replacement, but reordering head computation across stages (especially with GQA scheduling, where query heads are processed out of order) could introduce subtle differences in gradient accumulation due to floating-point non-associativity. Could you provide loss curves or validation perplexity over at least 1K steps showing UPipe matches standard Ulysses bit-for-bit (or within acceptable tolerance)?

* 1M sequence performance: UPipe shows lower throughput than baselines at sequence lengths ≤1M, which remain the most common regime for long-context workloads (SFT, RLHF, retrieval-augmented generation). Since UPipe's core benefit is freeing activation memory, it would strengthen the paper to show how this freed memory can be reinvested for net throughput gains—e.g., reducing activation checkpointing recomputation, increasing micro-batch size, enabling HSDP rather than FSDP. Without such a memory-for-speed tradeoff analysis, the practical value at ≤1M lengths remains unclear.

* Composability at scale: The experiments use FSDP + CP on at most 16 GPUs. In production settings, training typically combines TP + CP + DP (+ PP for larger models) with selective activation checkpointing. How does UPipe's stage-wise execution interact with tensor parallelism (where heads are already partitioned across TP ranks) and non-full activation checkpointing? Does the constraint U ≥ C still hold when C is the Ulysses subset of a hybrid TP×CP layout?

**Strengths And Weaknesses:**

###Strengths

S1. Drop-in replacement for DeepSpeed-Ulysses within TorchTitan.

S2. GQA scheduling design: reorders query heads to co-locate same-KV-group heads in one stage, avoiding redundant KV communication.

S3. Covers two model sizes (8B, 70B), two HW configs (8, 16 H100s), 128K–5M+, multiple baselines.

S4. Clear chunk-size tradeoff ablation.

### Weaknesses
W1. Incremental novelty. Headwise tiling follows naturally from the existing paradigm (tiled MLP, tiled loss → tiled attention heads). Conceptual leap is small.

W2. No convergence validation — no loss curves, perplexity, or downstream eval. Only throughput and memory. Numerical equivalence should be shown for a method that reorders attention computation.

W3. Scale limited to 16 GPUs. Multi-node bandwidth sensitivity unexplored. 70B experiments limited by CPU memory (OOCM at 1M).

W4. At sequence lengths ≤1M—which remain the most common long-context training regime—UPipe shows slightly lower throughput than standard Ulysses due to the overhead of repeated stage launches. The benefits only materialize at ≥2M, narrowing the method's practical applicability.

---

> ### Author Rebuttal · Authors · 2026-03-31
>
> We appreciate the thoughtful feedback from the reviewer. We have addressed the points raised by the reviewer below:
>
> **Q-1: Similarity with other tiling techniques**\
> R-1: We respectfully disagree. In tiled MLP or tiled CE loss, computation on one tile is independent of the other tiles. UPipe's headwise tiling does not share this property. Due to grouped query attention (GQA), one key head interacts with multiple query heads, and UPipe rearranges the attention computation to maximally realize memory benefits. The backward pass also requires out-of-order group-wise gradient accumulation and proactive dereferencing of Key/Value communication buffers to prevent memory leaks. These challenges are unique to headwise chunking, and UPipe provides a ready-to-use implementation for realizing these benefits.
>
> **Q-2: Training loss convergence**\
> R-2: We kindly request the reviewer to consult **R-3** in response to Reviewer **fwST**
>
> **Q-3: Multi-node bandwidth sensitivity**\
> R-3: Our multi-node results (Figure 4) demonstrate UPipe on 16 GPUs across 2 nodes using an 8-Ulysses-2-Ring hybrid setup. We also provide multi-node results for Qwen3-32B on 16xH100s in our response **R-4** to Reviewer **fwST**. As shown in **Table 3**, UPipe remains competitive across different context lengths on a multi-node setup. Since we use a USP-Hybrid style setup (section 5.2.2), all-to-all communication is restricted within the node, so UPipe does not introduce any additional inter-node traffic.
>
> **Q-4: Practical applications for <1M tokens / Repurposing the memory savings**\
> R-4: UPipe's memory savings remain valid for < 1M tokens while maintaining throughput on par with Ulysses when increasing the batch size. **Table 4** shows the throughput and memory comparison for context length = 128K tokens at different batch sizes on Llama3-8B using a single 8xH100 node. Note that at higher batch sizes, the time for CPU-GPU transfers over PCIe scales super-linearly (due to PCIe contention), leading to slight performance degradation.
>
> ### **Table 4**: UPipe throughput (TPS) and memory (GiB) at different batch sizes
>
> ||8|16|24|32|36|
> |-:|-|-|-|-|-|
> |**Ulysses (TPS)**|2381.85|2418.21|2390.20|OOM|OOM|
> |**UPipe (TPS)**|2394.37|2406.29|2374.01|2372.83|1678.61|
> |**Ulysses (GiB)**|33.86|48.40|62.94|OOM|OOM|
> |**UPipe (GiB)**|30.66|41.95|53.24|64.53|70.18|
>
> Following the reviewer's excellent suggestion, for batch size = 24, we repurpose UPipe's memory headroom to relax CPU activation offloading via strided offloading (skipping every 7th layer). As shown in **Table 5**, this yields better throughput than Ulysses. We also show repurposing memory savings towards selective activation checkpointing in R-6 below.
>
> ### **Table 5**: Repurposing UPipe memory savings with strided activation offloading
>
> |Seqlen=128K, BS = 24|Throughput (TPS)|Memory (GiB)|
> |-:|-|-|
> |**Ulysses**|2390.20|62.94|
> |**UPipe**|2374.01|**53.24**|
> |**UPipe w/ Strided-AO**|**2417.55**|62.39|
>
> **Q-5: Composability with other parallelism techniques**\
> R-5: UPipe is fully composable with Tensor, Data, and Pipeline Parallelism as a drop-in replacement of the Ulysses attention module. However, for long context training, TP offers no performance advantage and worse memory usage compared to FSDP + CP, since the memory savings from sharded weight matrices are outweighed by the all-gather overhead of the entire input sequence. Based on the white paper (section 3.4.2) from Megatron-Core [1] – a widely used distributed training framework, TP is only useful for large QKV projection matrices relative to the context length.
>
> For a TP x CP hybrid layout, each device processes a subset of attention heads independently. Each UPipe stage should process at least (C * TP) heads, giving the constraint U ≥ (C * TP), where C is the Ulysses context parallelism degree and TP is the Tensor Parallelism degree.
>
> [1] Megatron-Core: https://arxiv.org/pdf/2603.07685v1
>
> **Q-6: Compatibility with selective activation checkpointing**\
> R-6: UPipe's forward pass is implemented as a custom torch operator, so it natively supports PyTorch's selective activation checkpointing (SAC) and is compatible with Megatron-style checkpointing. We omit SAC in our experiments because TorchTitan retains layer inputs in addition to SAC-retained activations, increasing the memory usage. We focused on maximum memory efficiency in the paper, so we did not use SAC.
>
> **Table 6** compares Ulysses and UPipe with and without SAC on Llama3-8B (8xH100 node, 512K tokens). UPipe with SAC yields significantly higher throughput by avoiding attention recomputation in the backward pass, at a memory cost equal to the input buffer size (0.5GiB for 512K tokens, 5GiB for 5M tokens).
>
> ### **Table 6**: UPipe with selective activation checkpointing
>
> |Seqlen = 512K| Throughput (tokens/sec)|Memory (GiB)|
> |-:|:-:|:-:|
> |**Ulysses w/o SAC**|864.16|27.34|
> |**Ulysses w/ SAC**|**1066.01**|27.85|
> |**UPipe w/o SAC**|850.97|**25.24**|
> |**UPipe w/ SAC**|1056.77|25.75|

---

> > ### Author Rebuttal · Reviewer_c8BF · 2026-04-04
> >
> > W2 is resolved by the convergence validation (Table 2). W3 and W4 are improved with additional experiments.

---

> > > ### Author Response · Authors · 2026-04-04
> > >
> > > Thank you for acknowledging our rebuttal response! \
> > > Given our results address your questions and concerns, we would appreciate if you could readjust the score based on the final evaluation of our paper.

---

### Official Review · Reviewer_fwST · 2026-03-10

**Soundness:** 3
**Presentation:** 3
**Significance:** 3
**Originality:** 2
**Overall Recommendation:** 4
**Confidence:** 3

**Summary:**

This paper proposes UPipe (Untied Ulysses), a memory-efficient context parallelism method for long-context Transformer training. The key idea is to chunk attention computation along the head dimension, so that only a subset of heads is materialized and communicated at a time, reducing peak attention activation memory from dependence on the full number of heads to the chunk size. The paper also introduces a GQA-compatible scheduling strategy to avoid redundant KV communication, and combines the method with tiled FFN / cross-entropy kernels and activation checkpointing.

**Compliance With Llm Reviewing Policy:**

Affirmed.

**Key Questions For Authors:**

1. Can you provide a clearer ablation isolating UPipe alone, with all other memory-saving techniques held fixed?
2. Does the staged headwise execution have any effect on numerical behavior, convergence, or final model quality?
3. Why are the gains much clearer for 8B than for 70B?
4. How sensitive is the method to the head chunk size U?

**Limitations:**

See above.

**Strengths And Weaknesses:**

Strengths:
1. Targets an important practical bottleneck in long-context training.
2. The idea is simple, intuitive, and easy to understand.
3. The method is well motivated by memory analysis.
4. The design is compatible with modern architectures using GQA.

Weaknesses:
1. The novelty is somewhat incremental: the paper mainly proposes a more memory-efficient execution schedule for existing Ulysses-style context parallelism rather than a fundamentally new context parallel algorithm.
2. The evaluation does not fully isolate the contribution of UPipe itself: the overall system combines several memory-saving components, including tiled FFN, tiled CE loss, activation checkpointing, and CPU offloading, making it difficult to determine how much of the gain comes specifically from headwise chunking.
3. Final training quality is not deeply discussed: it would be useful to more explicitly confirm that the reordering does not affect optimization behavior, convergence.

---

> ### Author Rebuttal · Authors · 2026-03-30
>
> Thank you for your detailed feedback! We have addressed the points raised in the review below:
>
> **W1: Novelty of the headwise chunking technique**\
> R1: We kindly ask you to refer to **R1** in our response to Reviewer **c8BF** discussing the novelty of the chunking mechanism compared to prior tiling techniques.
>
> **Q1: Isolate the memory benefits of UPipe**\
> R2: Thank you for this question! Each of the memory optimizations mentioned in the paper (tiled MLP, tiled CE Loss, activation checkpointing, CPU offloading) are used for **_all_** experiments (all baselines and UPipe). Therefore, the results already show the isolated effect of UPipe on memory savings, which enabled larger context lengths.
>
> We provide the peak memory usage with different memory optimizations on top of Ulysses and UPipe in **Table 1**, using a single 8xH100 node with Llama3-8B and the context length of 512K tokens. We use tiled CE loss throughout to avoid running out of GPU memory. **AO** stands for activation checkpointing with CPU offloading.
>
> ### **Table 1**: Ablation of different memory optimization techniques
>
> |Memory (GiB)|**Vanilla**|**AO**|**TiledMLP**|**TiledMLP + AO**|
> |-:|:-:|:-:|:-:|:-:|
> |**Ulysses**|46.07|34.55|38.32|26.80|
> |**UPipe**|**45.35**|**33.80**|**36.23**|**24.70**|
>
> Without TiledMLP (columns 0 and 1), peak memory is dictated by the feed-forward stage, and the marginal memory saved by UPipe is due to better memory management in our implementation. TiledMLP + AO is necessary for training on multi-million context lengths, and UPipe's savings help achieve longer contexts. The memory savings scale linearly with context length (UPipe saves $\approx$2GiB for 512K, 4GiB for 1M, 8GiB for 2M, as shown in Table 4 of our paper).
>
> **Q2: Effect of UPipe on training loss convergence**\
> R3: UPipe is mathematically equivalent to Ulysses and performs the exact same computations, albeit in a different order. Therefore, Ulysses and UPipe might have minor differences in practice due to the non-associative nature of floating-point operations.
>
> For validating this, we train a Llama3-8B model using the **C4 dataset** [1] on a single 8xH100 node with randomly initialized weights, the learning rate of 3e-4 with linear decay, the batch size = 1, and the context length of 128K tokens. **Table 2** shows the training loss comparison between Ulysses and UPipe after every 100 steps up to 1000 steps. The difference remains roughly within 0.5% relative to Ulysses.
>
> ### **Table 2**: Training loss convergence over 1K steps
>
> |Steps|100|200|300|400|500|600|700|800|900|1000|
> |-|-|-|-|-|-|-|-|-|-|-|
> |**Ulysses**|6.83|6.58|5.98|5.62|5.22|4.88|4.94|5.11|4.68|4.47|
> |**UPipe**|6.85|6.54|5.98|5.62|5.22|4.90|4.95|5.11|4.69|4.47|
> |**%Diff**|0.29%|-0.61%|0.00%|0.00%|0.00%|0.41%|0.20%|0.00%|0.21%|0.00%|
>
> [1] https://huggingface.co/datasets/allenai/c4
>
> **Q3: Gains for larger model**\
> R4: The gains for 70B are primarily capped due to the CPU memory bottleneck. For Llama-70B, we used FSDP with CPU offloading to fit the model on a single 8xH100 node, causing significant memory pressure on CPU RAM.
>
> To illustrate UPipe's benefits for larger models on a multi-node setup, we ran Qwen3-32B [1] on 16xH100s (2 nodes). **Table 3** shows the throughput comparison. UPipe shows significant memory benefits for Qwen3-32B, supporting up to 4M tokens and outperforming FPDT in throughput across all context lengths.
>
> ### **Table 3**: Multi-node throughput comparison for Qwen3-32B
>
> |Throughput (TPS)|128K|256K|512K|1M|2M|3M|4M|
> |-:|:-:|:-:|:-:|:-:|:-:|:-:|:-:|
> |**Native PyTorch**|127.03|112.20|91.39|OOM|-|-|-|
> |**Ring**|418.39|308.88|194.44|110.27|58.45|OOM|-|
> |**Ulysses**|**545.29**|**370.70**|**217.04**|**117.02**|**59.98**|OOM|-|
> |**FPDT**|286.40|217.85|151.91|95.88|55.41|38.86|27.66|
> |**Ours**|483.29|339.56|204.46|113.26|**59.56**|**40.42**|**29.97**|
>
> [1] Qwen3-32B: https://arxiv.org/abs/2505.09388
>
> **Q4: Sensitivity to head chunk size U**\
> R5: We provide the sensitivity analysis of UPipe for a context length of 512K tokens with varying head chunk size U **in Section 5.4 and Figure 5 of our submission**. The head chunk size U allows for a smooth runtime-memory tradeoff, where a smaller chunk size yields better memory savings but slight performance degradation due to latency overheads from multiple stage launches. At longer context lengths, this overhead is amortized due to the larger duration of computation/communication kernels, so a smaller chunk size is optimal.

---

> > ### Author Rebuttal · Reviewer_fwST · 2026-04-03
> >
> > I have no further comments. Best of luck.

---

### Decision · Program_Chairs · 2026-04-30

**Decision:**

Accept (regular)

**Comment:**

This paper proposes a new context parallelism strategy that can substantially reduce peak memory usage and increase effective context length during training. Although some reviewers had initial concerns about limited novelty and practicality, all reviewers recommended acceptance after rebuttal and subsequent discussion. With all feedback from the rebuttal incorporated in the final revision, I recommend acceptance.